# Non-convex online learning via algorithmic equivalence

**Udaya Ghai** [*‡†]    **Zhou Lu**[*†]    **Elad Hazan**[*†]

## Abstract

We study an algorithmic equivalence technique between non-convex gradient descent and convex mirror descent. We start by looking at a harder problem of regret minimization in online non-convex optimization. We show that under certain geometric and smoothness conditions, online gradient descent applied to non-convex functions is an approximation of online mirror descent applied to convex functions under reparameterization. In continuous time, the gradient flow with this reparameterization was shown to be *exactly* equivalent to continuous-time mirror descent by Amid and Warmuth [4], but theory for the analogous discrete time algorithms is left as an open problem. We prove an $O(T^{\frac{2}{3}})$ regret bound for non-convex online gradient descent in this setting, answering this open problem. Our analysis is based on a new and simple algorithmic equivalence method.

## 1 Introduction

Gradient descent is probably the simplest and most popular algorithm in convex optimization, with numerous and extensive studies on its convergence properties. For non-convex objectives, it is known that gradient descent does not necessarily converge to the global optimum, which is in general NP hard. Thus recent research focuses on alternate objectives, such as finding first-order stationary points efficiently, or the importance of higher order stationary points, e.g. [19, 22, 1].

However, the study of global convergence of gradient descent for non-convex objectives is increasingly important due to the fact that in practice, gradient descent and its variants can achieve zero error on a highly non-convex loss function of a deep neural network.

To explain the success of modern deep learning it is thus important to understand why gradient descent can converge to a global minimum for some highly non-convex functions. Several important research directions have stemmed from this motivation, including the study of optimization for deep linear networks [20, 21, 6], non-convex matrix factorization [7], provable convergence for neural networks in the linearization regime of the neural tangent kernel [26, 12, 13, 2, 32, 18, 9], and more.

A recent promising approach, proposed in [4, 3], is to consider reparameterizing mirror descent as gradient descent. In particular, [4] shows that a continuous-time online mirror descent with a convex loss can be written in an equivalent form of a continuous-time online gradient descent whose loss isn't necessarily convex after reparameterization, but has significant structure. [3] further explores this idea, showing that after reparameterization, online gradient descent has the same worst-case regret bound as the exponentiated gradient algorithm. Both [4] and [3] provide a new prospect on understanding the convergence of non-convex gradient descent. More recently, [23] provides a more general result, tightly characterizing when gradient flow can be written as mirror flow by introducing the notion of a commuting parameterization.

---

[*]Google AI Princeton

[†]Princeton University

[‡]Correspondence to: <ughai@cs.princeton.edu>.

36th Conference on Neural Information Processing Systems (NeurIPS 2022).

## 1.1 The Amid-Warmuth question

The works of [4, 3] are limited in two respects: either they are restricted to the continuous-time gradient-flow setting, which is impractical, or they address specific algorithms (Winnow, Hedge, EG for linear regression) with a relative entropy divergence. Amid and Warmuth pose the question of extending the reparameterization idea to the general online convex optimization as well as the conditions where this can be obtained. Solving the question from [4] would give an answer to realistic scenarios and algorithms, with precise regret and running time bounds, for certain non-convex optimization problems that can be solved to global optimality using gradient descent.

## 1.2 Statement of results

Our study starts from the more general problem of regret minimization in online convex optimization. Specifically, we study the reparameterization of online mirror descent (OMD) as online gradient descent (OGD) [30] for general online convex optimization. We show that under certain geometric and smoothness conditions, a non-convex reparameterized OGD algorithm closely match the original convex OMD algorithm. Quantitative bounds on this matching allow us to prove that the OGD algorithm after reparameterization achieves an $O(T^{\frac{2}{3}})$ regret bound. This bound is discrete-time and fairly general, applying broadly to smooth, bounded mirror descent regularizers rather than anything more specific. We also provide sufficient conditions for non-convex losses such that OGD implicitly has an equivalent mirror descent regularizer and hence will have $O(T^{\frac{2}{3}})$ regret.

Our analysis relies on a simple and new algorithmic equivalence technique, which may be of independent interest. The key step is to show that the outputs of OMD and reparameterized OGD are very close to each other when initialized from the same point. Considering the OGD update as a perturbed version of OMD along with the fact that OMD can tolerate bounded (adversarial) perturbation per trial allows us to prove an $O(T^{\frac{2}{3}})$ regret bound for the OGD update.

This answers the open question raised in [4], generalizing from continuous-time gradient flow and specialized algorithms, to general online convex optimization. The regularization term of MD is also generalized from relative entropy to any strongly convex function. Further, for discrete-time reparameterization, our results extend to arbitrary Lipschitz convex loss functions, as opposed to custom analysis for specific settings of expert prediction, linear prediction, and linear regression [3].

## 1.3 Paper structure

Section 2 introduces the setting and sketches out background work in the continuous setting. Section 3 provides our main result, which is subsequently proven in Section 4. Section 5 provides analysis in the opposite direction, showing the existence of an equivalent mirror descent regularizer for some non-convex gradient descent problems. Some analysis is deferred to the Appendix.

## 1.4 Related works

Gradient descent is the simplest and one of the most popular algorithms in convex optimization. Extensive studies have been conducted to show the convergence of GD to the global minimizer in both the stochastic optimization setting [24], and the online convex optimization setting [31]. For the non-convex setting, however, finding the global minimum is NP-hard and most work focuses on the convergence to first-order stationary points instead. The well-known descent lemma guarantees the convergence of non-convex GD when the objective is smooth, and dropping smoothness is possible if other conditions are introduced [10]. Many works consider avoiding saddle points; [22] shows that GD converges to local minimizers a.s. if all saddle points are strict while [19] proposes perturbed gradient descent to escape saddle points. It's also shown that even in the online setting, chasing first-order stationary points is possible [17].

Mirror descent, first introduced by [25], generalizes the gradient descent method in the sense that it can adapt to the 'geometry' of the optimization problem [11]. Its analogue in the online setting, online mirror descent also achieves tight regret bounds [27] and without projection is shown to be equivalent to the classical regularized follow-the-leader (RFTL) method with constant step-size. Though it's natural to think of mirror descent as changing the geometry of the optimization objective in gradient

descent [15], the idea to explain non-convex GD by a corresponding convex MD is explored only recently.

The most relevant works to ours are [4] and [3], which prove the equivalence between continuous time OMD and OGD after reparameterization, and special discrete-time algorithms when the regularization is the relative entropy, respectively. Our results extend [4] and [3] to general discrete-time OCO with general regularization.

## 2 Preliminaries

### 2.1 Notation

For a function $f : \mathbb{R}^d \to \mathbb{R}^d$, we use the notation $J_f(x) : \mathbb{R}^d \to \mathbb{R}^d$ to denote the Jacobian of $f$ at $x$. We use the notaion $\odot$ to represents an element-wise product of vectors. Given a strictly convex function $R : \mathbb{R}^d \to \mathbb{R}^d$, we denote the Bregman divergence as

$$D_R(x\|y) := R(x) - R(y) - \nabla R(y)^\top (x - y) .$$

For a convex set $\mathcal{K}$ and strictly convex regularizer $R$, we use $\Pi_{\mathcal{K}}^R(x) := \arg\min_{y \in \mathcal{K}} D_R(y\|x)$ to denote Bregman projection, with shorthand $\Pi_{\mathcal{K}} := \Pi_{\mathcal{K}}^{\|\cdot\|^2}$ for Euclidean projection. Given a positive-definite matrix $M \in \mathbb{R}^{d \times d}$, we define the norm $\|x\|_M := \sqrt{x^\top M x}$. We use the notation, $B_p := \{x \in \mathbb{R}^d : \|x\|_p \le 1\}$ for an $\ell_p$ ball and $B_p^+$ to denote the intersection of the $\ell_p$ ball and the positive orthant.

### 2.2 Online convex optimization

We consider the online convex optimization (OCO) problem. At each round $t$ the player $\mathcal{A}$ chooses $x_t \in \mathcal{K}$ where $\mathcal{K} \subset \mathbb{R}^d$ is some convex domain, then the adversary reveals loss function $f_t(x)$ and player suffers loss $f_t(x_t)$. The goal is to minimize regret:

$$\text{Regret}(\mathcal{A}) = \sum_{t=1}^{T} f_t(x_t) - \min_{x \in \mathcal{K}} \sum_{t=1}^{T} f_t(x)$$

The player is allowed to get access to the (sub-)gradient $\nabla_x f_t(x_t)$ as well.

### 2.3 Reparameterizing continuous-time mirror flow as gradient flow

In the continuous-time setting, we have exact conditions from [4] where the trajectory of mirror-flow *exactly* coincides with that of gradient descent in a reparameterized space. Concretely, mirror flow on $f : \mathbb{R}^d \to \mathbb{R}$ is defined by the following ordinary differential equation (ODE):

$$\frac{\partial}{\partial t} \nabla R(x(t)) = -\eta \nabla f(x(t)) . \tag{1}$$

In Theorem 2 of [4], this update is shown to be equivalent to the following gradient flow ODE with $x(t) = q(u(t))$:

$$\frac{\partial u}{\partial t} = -\eta \nabla f(q(u(t))) , \tag{2}$$

if the mirror descent regularizer $R$ and reparameterization function $q$ satisfy $[\nabla^2 R(q(u))]^{-1} = J_q(u) J_q(u)^\top$. This relationship between the Hessian of the OMD regularizer and the reparameterization function assures that, up to second order factors in $\|u - v\|_2$

$$D_R(q(u)\|q(v)) \approx \frac{1}{2}\|u - v\|_2^2 , \tag{3}$$

so the geometry induced by $R$ is approximately transformed into a Euclidean geometry by $q^{-1}$. Because higher order factors vanish in continuous time, this assures that mirror flow and this reparameterized gradient flow coincide as desired.

## 2.4 Discretizing the updates

Unfortunately, continuous-time updates cannot be implemented in practice so we must rely on discretization. Using the forward Euler scheme for discretization of (1), yields the mirror descent algorithm. After discretization, the exact equivalence of mirror descent and reparameterized gradient descent no longer holds as higher order factors in (3) are relevant. This motivates the open problem of [4] of finding general conditions under which the discretizations of these continuous time trajectories closely track each other. We tackle extending these results through the online setting described above.

The Online Mirror Descent (OMD) algorithm has the following update on $x_t$:

$$\nabla R(y_{t+1}) = \nabla R(x_t) - \eta \nabla f_t(x_t)$$
$$x_{t+1} = \arg\min_{x \in \mathcal{K}} D_R(x \| y_{t+1})$$

where $R : \mathcal{K} \to \mathbb{R}$ is a 1-strongly convex regularization over the domain[4], $D_R$ is the Bregman divergence, and $y_1$ is initialized to satisfy $\nabla R(y_1) = 0$.

Another equivalent interpretation of OMD is

$$x_{t+1} = \arg\min_{x \in \mathcal{K}} \nabla f_t(x_t)^\top (x - x_t) + \frac{1}{\eta} D_R(x \| x_t) .$$

The most important special case is online gradient descent (OGD): $x_{t+1} = \Pi_{\mathcal{K}}(x_t - \eta \nabla f_t(x_t))$, which is OMD with $D_R(x \| x_t) = \frac{1}{2} \|x - x_t\|_2^2$. In this paper we consider reparameterizing general OMD update as a simple OGD update. We introduce assumptions of the regularizer, reparameterization, and losses in the sequel.

## 2.5 Assumptions

**Assumption 1.** *There exists a convex domain $\mathcal{K}' \subseteq \mathbb{R}^d$ and a bijective reparameterization function $q : \mathcal{K}' \to \mathcal{K}$ satisfying $[\nabla^2 R(x)]^{-1} = J_q(u) J_q(u)^\top$, with $x = q(u)$.*

We also make the following smoothness/Lipschitz assumptions:

**Assumption 2.** *Let $G > 1$ be a constant. We assume $q$ is $G$-Lipschitz, and the 1- strongly convex regularization $R$ is smooth with its first and third derivatives upper bounded by some constant $G$. The first and second derivatives of $q^{-1}$ are bounded by $G$. Furthermore, assume that for all $x \in \mathcal{K}$ $D_R(x \| \cdot)$ is $G$-Lipschitz over $\mathcal{K}$.*

Finally, we assume loss functions have bounded gradients and the convex domain has bounded diameter with respect to the Bregman divergence.

**Assumption 3.** *The loss functions $f_t$ have gradients bounded by $\|\nabla f_t(x)\|_2 \leq G_F$ for all $x \in \mathcal{K}$. The diameter of $\mathcal{K}$, $\sup_{x,y \in \mathcal{K}} D_R(x \| y) \leq D$ .*

## 3 Algorithm

We now provide our main result, a regret bound for Algorithm 2. We note, Algorithm 1 and Algorithm 2 are the (projected) forward euler discretizations of (1) and (2). It's tempting to directly analyze the regret of OGD with losses $\tilde{f}_t(u) = f_t(q(u))$. The main barrier in doing so is that $\tilde{f}$ isn't necessarily convex. However, we show that the OMD and OGD updates are in fact $O(\eta^{\frac{3}{2}})$-close to each other, therefore the OGD update still suffers only sublinear regret.

---

[4]Usually, the strong convexity of a mirror descent regularizer is with respect to some norm that is not necessarily the $\ell_2$ norm. In this work, the focus is on the dependence on the regret on $T$, so for simplicity we consider just the $\ell_2$ strong convexity, which will worsen constants, but does not affect the dependence on $T$.

| **Algorithm 1** Online Mirror Descent | **Algorithm 2** Online Gradient Descent |
|---|---|
| 1: Input: Initialization $x_1 \in \mathcal{K}$, regularizer $R$. | 1: Input: Initialization $u_1 \in \mathcal{K}' = q^{-1}(\mathcal{K})$. |
| 2: **for** $t = 1, \ldots, T$ **do** | 2: **for** $t = 1, \ldots, T$ **do** |
| 3:    Predict $x_t$ | 3:    Predict $u_t$ |
| 4:    Observe $\nabla f_t(x_t)$ | 4:    Observe $\nabla \tilde{f}_t(u_t) = \nabla f_t(q(u_t))$ |
| 5:    Update | 5:    Update |

$$y_{t+1} = (\nabla R)^{-1}(\nabla R(x_t) - \eta \nabla f_t(x_t))$$
$$x_{t+1} = \Pi_{\mathcal{K}}^R(y_{t+1})$$

$$v_{t+1} = u_t - \eta \nabla \tilde{f}_t(u_t)$$
$$u_{t+1} = \Pi_{\mathcal{K}'}(v_{t+1})$$

6: **end for**      6: **end for**

**Remark 1.** The gradient oracle $\nabla \tilde{f}_t$ can also be calculated from an oracle of $\nabla f_t$ if we know the reparameterization function $q$.

**Remark 2.** The reparameterized gradient descent uses Euclidean projection rather than Bregman projection. This can potentially be an advantage if Euclidean projection can be computed efficiently on $q^{-1}(\mathcal{K})$ (eg. via method like [29]), while Bregman projection is less efficient on $\mathcal{K}$.

**Theorem 3.** *Under Assumptions 1-3, by setting $\eta = T^{-2/3}D^{2/3}G^{-10/3}G_F^{-1}$ the regret of Algorithm 2 is upper bounded by $O(T^{2/3}D^{1/3}G^{10/3}G_F)$.*

Proof of the theorem is in Section 4.

### 3.1 Examples

We now provide a few example settings. Note that in all three settings, $q$ are such that $\tilde{f}(u) = f(q(u))$ can be nonconvex for a convex $f$.

**Exponentiated gradient using quadratic reparameterization** One of the most popular algorithms for online learning over the filled-in simplex[5] $\mathcal{K} = B_1^+$ is the Exponentiated Gradient (EG) [8]. In this case, we show that OGD with quadratic reparameterization $q(u) = \frac{1}{4}u \odot u$ has vanishing average regret. We empirically verify that reparameterized GD iterates and EG iterates stay close on a toy problem in Figure 1. In [3] custom analysis is provided for linear losses that achieves optimal regret, but our approach has a simpler and more general analysis. If $R(x) = \sum_{i=1}^d x_i \log(x_i)$ then OMD is EG. In this case, if $x = q(u)$, then

$$J_q(u)J_q(u)^\top = \text{diag}(u/2)\text{diag}(u/2) = \text{diag}(u \odot u/4) = \text{diag}(q(u)) = \text{diag}(x) = [\nabla^2 R(x)]^{-1}$$

$$K' = q^{-1}(B_1^+) = \{u \in \mathbb{R}_+^d : \frac{1}{4}\sum_{i=1}^d u_i^2 \leq 1\} = 2B_2^+.$$

We note that for this example, Assumptions 2 will not hold as relative entropy is not sufficiently behaved near the boundary of the simplex. This can be handled by modifying $\mathcal{K}$ to be the smoothed simplex where all weights are at least $\varepsilon$ for some $\varepsilon$. This is a fairly standard technique. $\varepsilon$ can be chosen such that the full regret in sublinear (see App. B).

**Log barrier with exponential reparameterization** Consider the case of a log-barrier regularization $R(x) = -\sum_{i=1}^d \log(x_i)$ with $\mathcal{K} = [\varepsilon, 1]^d$. It can readily be shown that the box constraint maps to a box constraint. Consider $x = q(u) = \exp(u)$, where the exponential is elementwise. We see Assumption 1 is satisfied as

$$J_q(u)J_q(u)^\top = \text{diag}(\exp(u))\text{diag}(\exp(u)) = \text{diag}(q(u)^2) = \text{diag}(x^2) = [\nabla^2 R(x)]^{-1} .$$

**Tempered Bregman divergences with power reparameterization** A more recently studied family of mirror descent regularizers interpolate between $\ell_2$ regularization and negative entropy regularization [5]. Regularization for this family uses link function $\nabla R(x) = \log_\tau(x) = \frac{1}{1-\tau}(x^{1-\tau} -$

---

[5]One drawback of our approach is that the current analysis does not work for the true simplex like [3] because $q^{-1}(\mathcal{K})$ is the positive part of the unit sphere, which is not a convex set.

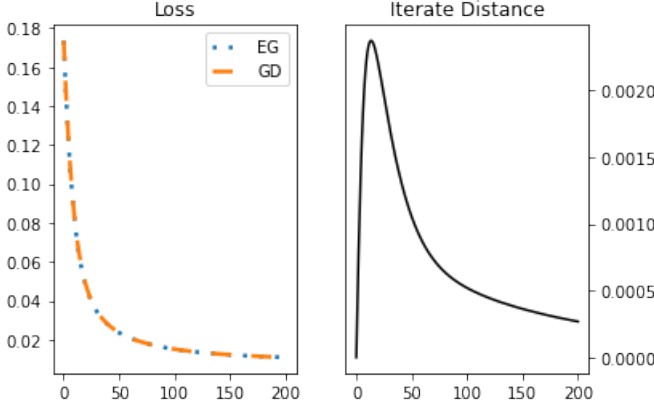

Figure 1: Gradient Descent using reparameterization $q(u) = \frac{1}{4} u \odot u$ produces iterates that closely track Exponential Gradient for a simple fixed quadratic loss example.

1), where $\tau$ is a *temperature* that can range between 0 and 1. As $\tau$ approaches 1, the link function approaches the natural logarithm, and hence the update approaches EG, while $\tau = 0$ corresponds to $\ell_2$ regularization. Such divergences are amenable to reparameterization when $\mathcal{K} = B_p^+$.

To see this, ignoring constant factors $[\nabla^2 R(x)]^{-1} = x^\tau$ for $\tau \in [0, 1)$. Now if $q(u) = u^{\frac{2}{2-\tau}}$ where the power is elementwise, then $J_q(u) = \text{diag}(u^{\frac{\tau}{2-\tau}}) = \text{diag}(q(u)^{\tau/2}) = \text{diag}(x^{\tau/2})$ as desired. Now, we note that the reparameterization is a power, so $B_p^+$ would get mapped to the $B_{\frac{2p}{2-\tau}}^+$, which is convex as long as $p > 1 - \tau/2$.

## 3.2 Challenges

In some situations, we do not have a closed form solution to the differential equation described by $J_q$. For example with the $\beta$-hypentropy regularizer of [14], the reparameterization involves $q^{-1}(x) = \int (x^2 + \beta^2)^{-1/4} dx$, and this integral does not have a known closed form. Furthermore, other cases of interest, like von Neumann Entropy regularization for Matrix Multiplicative Weights [28] seem difficult to fit into this framework as Assumption 1 is challenging to use with spectral functions.

## 4 Reparameterization analysis

We now move on to the proof of Theorem 3. The proof idea is to show that the OMD and OGD iterates are close to each other after a single update step starting from the same initial point. Then we can view the OGD update a perturbed version of OMD, and combine it with the fact that the OMD algorithm can tolerate bounded noise per trial.

We begin with the following key lemma showing that the updates $x_{t+1}$ and $q(u_{t+1})$ created by OMD and OGD respectively, are close to each other from the same initial point $x_t = q(u_t)$. To do this, we carefully analyze the errors that occur due to the approximation in (3) starting from a proximal formulation of mirror descent. We can show that the two updates are solutions to approximately the same strongly-convex objective, hence the solutions must be close.

**Lemma 4.** *Suppose Assumptions 1-3 hold and $x_t = q(u_t)$, then we have that $\|x_{t+1} - q(u_{t+1})\|_2 = O(G^4 G_F^{3/2} \eta^{3/2})$.*

*Proof.* We consider the following forms of Algorithms 1 and 2.

$$x_{t+1} = \arg\min_{x \in \mathcal{K}} \nabla f_t(x_t)^\top (x - x_t) + \frac{1}{\eta} D_R(x || x_t)$$

$$u_{t+1} = \arg\min_{u \in \mathcal{K}'} \nabla \tilde{f}_t(u_t)^\top (u - u_t) + \frac{1}{2\eta} \|u - u_t\|_2^2$$

We observe that both objectives we are minimizing can be written as the sum of a linear function plus a strongly convex function. In particular, if $\|x - x_t\|_2 > 2\eta G_F$, then the objective takes positive value which can't be minimal because taking $x = x_t$ gives 0. Define $\mathcal{K}_{r,x_t} = \mathcal{K} \cap \{x : \|x - x_t\|_2 \leq r\}$, then the OMD iterate $x_{t+1} \in \mathcal{K}_t := \mathcal{K}_{2\eta G_F, x_t}$. Likewise, $u_{t+1} \in \mathcal{K}'_t := \mathcal{K}'_{2\eta G_F \cdot G, u_t}$.

Using Taylor expansion with Assumption 2, for $D_R(x || x_t)$ with $\|x - x_t\|_2 \leq 2\eta G_F$, we rewrite it as

$$D_R(x || x_t) = R(x) - R(x_t) - \nabla R(x_t)^\top (x - x_t)$$
$$= \frac{1}{2}(x - x_t)^\top \nabla^2 R(x_t)(x - x_t) + O(G \cdot G_F^3 \eta^3)$$
$$= \frac{1}{2}\|x - x_t\|_{\nabla^2 R(x_t)}^2 + \varepsilon(x) \,,$$

where $\varepsilon(x) = O(G \cdot G_F^3 \eta^3)$ for $x \in \mathcal{K}_t$. And the OMD update becomes

$$x_{t+1} = \arg\min_{x \in \mathcal{K}_t} \nabla f_t(x_t)^\top (x - x_t) + \frac{1}{2\eta}\|x - x_t\|_{\nabla^2 R(x_t)}^2 + \varepsilon(x) \,.$$

We now use Taylor expansion to relate $\nabla \tilde{f}_t(u_t)^\top (u - u_t)$ to $\nabla f_t(x_t)^\top (x - x_t)$. Let $x = q(u)$ for $u \in \mathcal{K}'_t$, hence $x \in \bar{\mathcal{K}}_t := \mathcal{K}_{2\eta G_F \cdot G^2, x_t}$, then we have

$$\nabla \tilde{f}_t^\top (u_t)^\top (u - u_t) = \nabla f_t(x_t)^\top J_q(u_t)(u - u_t)$$
$$= \nabla f_t(x_t)^\top J_q(u_t)(q^{-1}(x) - q^{-1}(x_t))$$
$$= \nabla f_t(x_t)^\top J_q(u_t)(J_q^{-1}(u_t)(x - x_t)) + \tilde{\varepsilon}(x)$$
$$= \nabla f_t(x_t)^\top (x - x_t) + \tilde{\varepsilon}(x) \,,$$

where $\tilde{\varepsilon}(x) = O(G^5 G_F^3 \eta^2)$.

Using the above approximation, we can rewrite the OGD update as

$$q(u_{t+1}) = \arg\min_{x \in \bar{\mathcal{K}}_t} \nabla f_t(x_t)^\top (x - x_t) + \frac{1}{2\eta}\|q^{-1}(x) - q^{-1}(x_t)\|_2^2 + \tilde{\varepsilon}(x) \,.$$

On the other hand, we can rewrite $\frac{1}{2}\|q^{-1}(x) - q^{-1}(x_t)\|_2^2$ as

$$\frac{1}{2}\|q^{-1}(x) - q^{-1}(x_t)\|_2^2 = \frac{1}{2}\|J_q^{-1}(x_t)(x - x_t) + O(G^5 G_F^2 \eta^2)\|_2^2$$
$$= \frac{1}{2}\|x - x_t\|_{J_q^{-1}(x_t) J_q^{-1}(x_t)^\top}^2 + O(G^8 G_F^3 \eta^3)$$

Then by Assumption 1, $\nabla^2 R(x_t) = J_q^{-1}(x_t) J_q^{-1}(x_t)^\top$, so we conclude that $\frac{1}{\eta} B_R(x || x_t)$ and $\frac{1}{2\eta}\|q^{-1}(x) - q^{-1}(x_t)\|_2^2$ are $O(G^8 G_F^3 \eta^2)$-close. We can then rewrite the OGD update as

$$q(u_{t+1}) = \arg\min_{x \in \bar{\mathcal{K}}_t} \nabla f_t(x_t)^\top (x - x_t) + \frac{1}{2\eta}\|x - x_t\|_{\nabla^2 R(x_t)}^2 + \bar{\varepsilon}(x) \,,$$

where $\bar{\varepsilon}(x) = O(G^8 G_F^3 \eta^2)$ for $x \in \bar{\mathcal{K}}_t$. We note that $\mathcal{K}_t \subseteq \bar{\mathcal{K}}_t$ as needed.

Comparing both OMD and OGD updates, the objective functions are only off by an $O(G^8 G_F^3 \eta^2)$ term, therefore the minimizers are also $O(G^4 G_F^{3/2} \eta^{\frac{3}{2}})$-close since the objective functions are both $\Theta(\frac{1}{\eta})$ strongly convex. $\qquad\square$

**Remark 5.** It can be shown that if projection is not required, $\|x_{t+1} - q_{u_{t+1}}\|_2 = O(\eta^2)$ in Lemma 4. Furthermore, if the projection operation that Algorithms 1 and 2 perform are the same, the distance only gets closer. Such a bound would result in an $O(\sqrt{T})$ regret bound. For the case of EG over the simplex, the projection operations in both are the same weight scaling operations, so this could be a part of making $O(\sqrt{T})$ regret possible, though this still requires a better approach for handling exploding Lipschitz constants (see App. B).

Next, we show that the OMD algorithm is actually robust to noise: with a (potentially non-stochastic) bounded perturbation on the output $x_t$ per round, we can still get vanishing regret.

**Lemma 6.** *Suppose Assumptions 2 and 3 hold and Algorithm $\mathcal{A}$ does the following update:*

$$x_{t+1} = r_t + \arg\min_{x \in \mathcal{K}} \nabla f_t(x_t)^\top (x - x_t) + \frac{1}{\eta} D_R(x \| x_t)$$

*where $\|r_t\|_2 \leq C$. The regret of Algorithm $\mathcal{A}$ can be upper bounded by*

$$Regret(\mathcal{A}) \leq \frac{CTG}{\eta} + \frac{D}{\eta} + \frac{\eta G^2 T}{2}$$

This follows from a slightly modified analysis of OMD from [16], where the effect of the perturbation is bounded due to the Bregman divergence being Lipschitz. In particular, we end up with the standard regret bound for mirror descent that follows from a telescoping argument with an additional error term

$$\frac{1}{\eta} \sum_{t=1}^{T} D_R(x^* \| x_t) - D_R(x^* \| x_t^R) \leq \frac{CTG}{\eta} \;,$$

where $x_t^R$ is the counterfactual mirror descent iterate with $r_t = 0$.

Full proof of Lemma 6 can be found in Appendix A. Combining both lemmas, we complete our proof of Theorem 3:

*Proof.* Algorithm 2 gives a perturbed version of OMD with perturbation bounded by $C = O(\eta^{\frac{3}{2}})$ due to Lemma 4. Plugging $C = O(G^4 G_F^{3/2} \eta^{3/2})$ into the bound of Lemma 6, the regret is upper bounded by $O(G^5 G_F^{3/2} \sqrt{\eta} T + \frac{D}{\eta})$ ignoring the lower order term. Optimizing this and all constants gives the choice of $\eta = T^{-2/3} D^{2/3} G^{-10/3} G_F^{-1}$ and regret bound $O(T^{2/3} D^{1/3} G^{10/3} G_F)$. □

## 5 Implicit OMD reparameterization

We have shown that a general convex OMD can be reparameterized as a (potentially) non-convex OGD, with vanishing $O(T^{\frac{2}{3}})$ regret. The other direction from OGD to OMD is even more interesting: given a non-convex OGD, can we show its global convergence by showing the existence of a convex OMD which corresponds to OGD implicitly?

In other words, given a convex and compact domain $\mathcal{K}'$, what do we need to know about the domain $\mathcal{K}'$ and the (not necessarily convex) loss $\tilde{f}_t$ to ensure the existence of such equivalence? Fully characterizing the necessary and sufficient condition seems hard, and we provide here a simple sufficient condition which covers some known scenarios.

**Assumption 4.** *We assume the following properties about q.*

- *There exists a function $q$ such that $\tilde{f}_t(u)$ can be written as $f_t(q(u))$ where $f_t$ is convex.*

- *$q$ is a $C^3$-diffeomorphism, and $J_q(u)$ is diagonal.*

- *$q(\mathcal{K}')$ is convex and compact.*

We argue that once the conditions above are satisfied, there exists a regularization $R$ and a corresponding OMD with the desired equivalence. The only thing we need to verify is the existence of a strongly convex regularization $R$ which satisfies Assumption 1. We first show that $J_q(u)$ always has

non-zero (in fact, with the same sign because of continuity) determinant due to the fact that $q$ is a diffeomorphism. Then $R$ can be reconstructed by integrating twice its Hessians which we compute from the equation in Assumption 1. $R$ is also strongly convex as the Hessians are strictly PSD. We note that such conditions may not be tight, but are general, succinct and may cover some interesting examples.

**Theorem 7.** *Given a convex and compact domain $\mathcal{K}'$, and not necessarily convex loss $\tilde{f}_t$ satisfying Assumption 3. When Assumption 4 is met, there exists an OMD object with convex loss $f_t$, a convex domain and a strongly convex regularization $R$ satisfying Assumption 1. As a result, running Algorithm 2 on loss $\tilde{f}_t(u)$ has regret upper bound $\tilde{O}(T^{\frac{2}{3}})$.*

*Proof.* The first two properties are satisfied by the assumption. We verify the third property by constructing a regularization $R$, which is strongly convex and satisfies Assumption 1. The rest follows Theorem 3. By the fact that $q$ is a diffeomorphism, $J_q(u)$ is an invertible matrix. The fact that $J_q(u)$ is an invertible matrix implies that $J_q(u)J_q(u)^\top$ is invertible, and further $J_q(u)J_q(u)^\top \succ 0$.

Denote $H(u) = [J_q(u)J_q(u)^\top]^{-1} \succ 0$ which is well defined by the above argument, we want to construct $R$ such that $\nabla R^2(q(u)) = H(u)$. By the assumption that $J_q(u)$ is diagonal, we can write $q(u)$ as

$$q(u) = (q_1(u_1), q_2(u_2), ..., q_d(u_d))$$

where each $q_i$ is a scalar function, such that $H(u)$ is diagonal with $H(u)_{ii} = 1/(q_i'(u_i))^2$, which only depends on $u_i$. We set $R(q(u)) = \sum_{i=1}^d R_i(q_i(u_i)))$ and denote the function $\tilde{R}_i(u_i) = R_i(q_i(u_i))$ with variable $u_i$, then by the chain rule we have that

$$\tilde{R}_i{}'(u_i) = q_i'(u_i)R_i'(x_i)$$

$$\tilde{R}_i{}''(u_i) = q_i''(u_i)R_i'(x_i) + (q_i'(u_i))^2 R_i''(x_i) \ .$$

Plugging the first formula into the second one, we eliminate the $R_i'(x_i)$ term:

$$\tilde{R}_i{}''(u_i) = \frac{q_i''(u_i)\tilde{R}_i{}'(u_i)}{q_i'(u_i)} + (q_i'(u_i))^2 R_i''(x_i) \ .$$

Recall, we want to find $R$ such that $R_i''(x_i) = 1/(q_i'(u_i))^2$. Plugging it back to the equation above, the problem is reduced to find $R_i$ such that it satisfies the ODE

$$q_i'(u_i)\tilde{R}_i{}''(u_i) - q_i''(u_i)\tilde{R}_i{}'(u_i) - q_i'(u_i) = 0 \ .$$

It's known that there exists a solution to any linear and continuous second-order ODE. In fact, using the standard variation of constant method, one example solution $\tilde{R}_i{}'(u_i)$ can be

$$\tilde{R}_i{}'(u_i) = q_i'(u_i) \int_{C_i}^{u_i} \frac{1}{q_i'(u)} du + c_i q_i'(u_i) \ .$$

where $c_i, C_i$ are constants and $C_i = \min_{u \in \mathcal{K}'} u_i$ . It's well-defined because $q_i'(u_i)$ always has the same sign. To verify the correctness, we calculate that

$$\tilde{R}_i{}''(u_i) = q_i''(u_i) \int_{C_i}^{u_i} \frac{1}{q_i'(u)} du + 1 + c_i q_i''(u_i) \ ,$$

which solves the ODE together with the solution of $\tilde{R}_i{}'(u_i)$.

Next, plugging $\tilde{R}_i(u_i) = \int_{C_i}^{u_i} \tilde{R}_i{}'(u)du$ and $u = q^{-1}(x)$ into the expression of $R$ gives a solution of the regularization. Its strong-convexity is implied by the fact that $\nabla_R^2(x) = H(u) \succ 0$ and is continuous over a compact domain, thus there exists a constant $c > 0$ such that $\nabla_R^2(x) \succ cI$.

Finally, $q$ being a $C^3$-diffeomorphism means that both $q$ and $q^{-1}$ are 3 times continuously differentiable. By the compactness of domains and the continuity of the derivatives, there exists a constant $G$ satisfying Assumption 2. A similar argument holds for $R$. $\qquad \square$

# 6 Conclusion

We study the convergence of non-convex gradient descent in this paper, by showing (approximate) algorithmic equivalence between gradient descent and mirror descent in the discrete setting. We prove that under certain geometric and smoothness conditions, running OGD on non-convex losses obtained by reparameterizing a convex OMD has regret bound $O(T^{\frac{2}{3}})$, answering an open problem in [4]. Our analysis is based on a new algorithmic equivalence technique, combining the one-step closeness between OMD and the reparameterized OGD and the robustness of OMD. We further extend our result to the other direction, providing sufficient conditions for a non-convex OGD to have an implicit corresponding OMD and thus converge well. We leave several questions as future directions.

1. Is the $O(T^{\frac{2}{3}})$ regret bound improvable to $O(\sqrt{T})$ in general? The decay in the bound in this work seems to be caused by differences in projection in the reparameterized space and the original Bregman projection. Perhaps a different analysis technique can produce optimal bounds.

2. Can assumptions for reparameterization be relaxed? For example, it is not clear that Assumption 1 is a necessary condition. In the implicit mirror descent direction, it may be possible to lift the diagonal $J_q(u)$ assumption in Theorem 7 with more refined conditions on PDEs. Perhaps the commuting parameterization conditions from [23] can be adapted from the continuous to the discrete setting.

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
