# OpenReview forum: "Non-convex online learning via algorithmic equivalence"
_NeurIPS.cc/2022/Conference — NeurIPS 2022 Accept_

### Official Review · Reviewer_ARJZ · 2022-07-02

**Rating:** 5
**Confidence:** 3
**Soundness:** 3 good
**Presentation:** 3 good
**Contribution:** 2 fair

**Summary:**

This paper shows that for a certain class of non-convex problems, online gradient descent has a O(T^{2/3}) regret. Previous works by Amid and Warmuth show that gradient flow for minimizing a reparametrized function is equivalent to mirror flow for minimizing the original function. This work takes a step forward by showing in discrete-time, online gradient descent on a reparametrized objective has a sublinear regret.

The high-level idea is that the discrete-time online mirror descent has a O(\sqrt{T}) for the same class of problems. By carefully bounding the discrepancy between online mirror descent for the original problem and online gradient descent (OGD) for the non-convex re-parametrization, a sublinear regret of OGD can be obtained.




**Questions:**

N/A

**Strengths And Weaknesses:**

While the corresponding continuous-time result has been established in the previous work, the discrete-time setting seems to be left open. This work fills the gap.

In my opinion, this paper is pretty much borderline for the following reasons:

1. While I appreciate the theoretical result, I think it will be more helpful if the authors can discuss the advantage of conducting discrete-time online learning on the reprametrized functions, since the regret gets worse, i.e., one could simply use OMD for the convex problems.

2. Following the first point, I think the title should be modified and be more specific, since the technique seems only applicable to a certain class of problems and critically relies on the particular connection between OGD and OMD that has been established in the prior work by Amid and Warmuth.




Minor:
1. I don't quite understand Line 200-202. Perhaps a more detailed derivation would be helpful?
2. Line 215: "both both".

---

> ### Author Response · Authors · 2022-08-02
> **Response**
>
> We thank the reviewer for the valuable feedback.
>
> Weaknesses:
>
> 1.  The main benefit is to use the equivalence to identify whether a certain non-convex OGD problem converges without knowing its OMD counterpart. A preliminary result is shown in Theorem 7. Also, OGD uses Euclidean projection, which is sometimes easier to implement than Bregman projection used by OMD.
>
> 2. Yes we agree that our results heavily depend on the reparameterization viewpoint. The novelty of our paper is the use of algorithmic equivalence as a technique so we feel this is appropriate, even though our result does not hold for general nonconvex losses (which is NP Hard).
>
> Thank you for catching the typo on line 215, we have fixed this.  As for line 200-202, strong convexity assures us that the minimizer must be close to $x_t$ so we restrict to a sufficiently small ball around $x_t$ in order to use Taylor expansion arguments downstream. The notation is indeed a bit messy, but we are not sure how this argument should be expanded for improved clarity.

---

> > ### Comment · Reviewer_ARJZ · 2022-08-09
> > **Comment**
> >
> > I have checked the comments of Reviewer x2pL and Reviewer NwoT. As they bring up some issues that I didn't raise, e.g., the issue of G raised by Reviewer NwoT, I tend to keep the score as is.
> >
> > In my opinion, the title of the paper sounds grandiose, but the result (a bit) falls short of it.

---

### Official Review · Reviewer_NwoT · 2022-07-08

**Rating:** 6
**Confidence:** 5
**Soundness:** 2 fair
**Presentation:** 3 good
**Contribution:** 3 good

**Summary:**

The paper studies an approximate equivalence between online gradient
descent (OGD) on non-convex losses with online mirror descent (OMD) on
convex reparametrizations of the losses. Special cases of this
equivalence were already known from [3] and [4], but the present paper
attempts to set up a general theory. Its main results are:
- Theorem 3, which gives sufficient general conditions for the
  approximate equivalence to go through, leading to an O(T^{2/3}) regret
  bound for OGD on the non-convex losses.
- Section 3.1, which provides example instances of OMD on convex losses
  which can be approximated this way by OGD on non-convex losses.
- Theorem 7 goes in the other direction: it gives sufficient conditions
  for OGD on non-convex losses to be approximated by OMD on convex
  losses.



**Questions:**

* Could you indicate how the G that appears in Theorem 3 scales with T,
  and verify that R is 1-strongly convex for all three examples in
  Section 3.1?
* Could you answer the remark above about the proof of Theorem 7?



**Limitations:**

Section 3.2 provides a very good discussion of some of the limitations
of the approach.

The limitations in footnotes 1 and 2 seem sufficiently important to put
them somewhere in the main text.


**Strengths And Weaknesses:**

Strengths:
* The paper is very well written, especially the introduction.
* The approximate equivalence between non-convex and convex online
  optimization is an exciting new direction, for which few results are
  yet available. It would be great to have a general theory that
  describes when such equivalences are possible.

Weaknesses:
* It is not clear that the general theory covers any interesting cases:
  1. It does not recover the previous results from [3] and [4].
  2. I do not consider the examples provided in Section 3.1 to be very
  convincing. In the first two examples, it would seem that G could be
  as large as T, so then Theorem 3 would be vacuous. For the last
  example, I am not sure how the constants play out.
  3. Theorem 7 does not quantify how the strong convexity parameter c>0
  of the regularizer R that is constructed, depends on the properties of
  q and tilde{f}. It could therefore be

In spite of these weaknesses, the overall proof approach does seem like
the natural way forward, by quantifying the approximation error in the
equivalence. If my questions below can be answered satisfactorily, I am
therefore still mildly in favor of acceptance, even if there are no
strong examples.

Remarks:
* Theorem 3 assumes that R is 1-strongly convex. This should be stated
  explicitly in its conditions. I am also concerned that this is not
  verified in the examples from Section 3.1, so do they really satisfy
  this condition?
* In the Proof of Theorem 7: I don't understand how you get from the
  displayed equations above "solving the above formulas" to the ODE
  below.

Minor Comments:
* Line 80: equivalence between OMD and FTRL requires constant step size.
* "Euclidean" should be capitalized throughout the paper
* Line 95: p should be a subscript in p-norm
* The square on the Hessian appears to overlap with the nabla symbol
  throughout the paper.
* Line 108: Shouldn't q^{-1} be q? (Maybe I was just confused here.)
* Assumption 1: Should the Hessian be interpreted as the Hessian of R or
  the Hessian of R(q(.))
* Algorithm 1: f_t -> f_t(x_t)
* Line 147: "cab" -> "can"
* Line 205: f_t -> f_t(x_t)

---

> ### Author Response · Authors · 2022-08-02
> **Response**
>
> We thank the reviewer for the detailed comments, and will address them in order below.
>
> Weaknesses:
>
> 1. Our algorithm is designed to cover the general case, while the algorithms in [3],[4] are more customized. For example, the algorithms in [3],[4] do normalization instead of projection. We can still get sub-linear regret bounds for the same setting, see the answer to (2).
>
> 2. Good point! Thank you for noticing the potential issue on constants which was neglected. Our analysis can still get sub-linear regret bounds by smoothing the domain: we use a $T^{-\epsilon}$ smoothing instead in lines 163-164. Basically $G$ is dominated by the 3-nd derivative of $R$ which is roughly $T^{2\epsilon}$, and the regret can be bounded by $O(T^{1-\epsilon}+TG^5 \eta^{1/2}+1/\eta)$. Balancing the parameters by setting $\eta=T^{-20\epsilon/3-2/3}$ and then $\epsilon=1/23$ gives an $O(T^{22/23})$ regret bound. Though our general analysis can't recover optimal regret bounds for each case, it does preserve a sub-linear regret. We have added a proof to this in appendix in the latest version.
>
> 3. Theorem 7 is a general proof of existence and doesn't focus on the exact regret bound, therefore we don't require "constructing" these constants explicitly. But for certain cases like example 1 we do need to pay more attention.
>
>
> Question:
>
> 1. If we use a $T^{-\epsilon}$ smoothing in lines 163-164, $G$ is dominated by the 3-nd derivative of $R$ which is roughly $T^{2\epsilon}$.
> 2. Indeed we omitted the use of Assumption 1, which gives an additional formula $(q'_i(u_i)^2)R''_i(x_i)=1$. Inserting this and formula 1 into formula 2, we get rid of the dependence on $R'_i(x_i)$ in formula 2: $\tilde{R_i}''(u_i)-q_i''(u_i)\tilde{R_i}'(u_i)/q_i'(u_i)-1=0$. Because $q_i'(u_i)$ is positive we reach the final ODE. We have updated these intermediate steps in the latest version.
>
> Remarks:  We have added $1$-strongly convex to Assumption $2$. We assumed this for simplicity, as a strongly convex regularizer can be scaled by a constant to be $1$-strongly convex.  All three examples we use are $1$-strongly convex though, which can be confirmed by taking Hessians.  The Hessians are all diagonal so a simple minimization will show the result.
>
> Minor Comments: We have fixed the typos in the most recent revision. Thank you for pointing these out. For line 108, the $K$ is transformed via $q^{-1}$ into $K'$.

---

> > ### Comment · Reviewer_NwoT · 2022-08-08
> > **Reply to the authors' response**
> >
> > I would like to thank the authors for their clear answers to the weaknesses and questions I posed.
> >
> > I agree with their answers, but I still believe that weakness 1 is a limitation of the work, and it is also unfortunate that the answer to weakness 2 leads to an $O(T^{22/23})$ rate. Together these imply that the generality of the proposed framework does come at the cost of significantly deteriorated rates. I understand the authors' point that having any sublinear rate at all is a step forward, but it still seems that some key aspect is missing from the approach if it does not give good rates for any natural example. I will therefore keep my original score.

---

### Official Review · Reviewer_x2pL · 2022-07-11

**Rating:** 5
**Confidence:** 4
**Soundness:** 3 good
**Presentation:** 3 good
**Contribution:** 2 fair

**Summary:**

This paper shows that under several assumptions, online mirror descent for convex losses can be written as a perturbed version of online gradient descent for possibly non-convex losses with an $O( T^{2/3} )$ regret, up to a reparameterization. By a similar understanding, this paper also discusses when online gradient descent for non-convex losses is provably no-regret by the reparameterization argument.

**Questions:**

1. Please check the correctness of the first example in Section 3.1.
2. Is there any relevant application of Theorem 7 in Section 5?

**Limitations:**

Assumption 2 basically assumes that everything should be Lipschitz. This assumption is violated for exponentiated gradient descent and online mirror descent with self-concordant barriers, for example. The authors also noticed that the matrix version of exponentiated gradient "is difficult to fit into this framework." The self concordance of the regularizers in these examples may a useful property to replace the Lipschitz assumption.

**Strengths And Weaknesses:**

**Strengths**
1. The argument is very simple yet reasonable and actually novel, as far as I know.

**Weaknesses**
1. The first example in Section 3.1 seems to be wrong. That example considers online exponentiated gradient descent on the probability simplex, arguably the most representative instance of online mirror descent. As relative entropy is not Lipschitz on the probability simplex, violating Assumption 2, the smoothed simplex is proposed to be a fix. However, then points close enough to the origin should be removed from the set $\mathcal{K}'$, meaning that $\mathcal{K}'$ is non-convex and the theory in the paper does not apply.
2. The other two examples in Section 3.1, online mirror descent with the log barrier and the tempered Bregman divergence, are much less representative compared to exponentiated gradient descent.
2. Section 5, which provides conditions when online gradient on non-convex losses may be viewed as online mirror descent on convex losses and hence is no-regret, should be the most useful part. Unfortunately, no relevant example is provided.
2. There are some typos.
    - Ln. 95: $\Vert x \Vert_p$ instead of $\Vert x \Vert p$
    - Ln. 105: $\nabla^2 R$ instead of $\nabla^2_R$
    - The equality following Ln. 205: $\nabla f_t ( x_t )$ instead of $\nabla f_t$
    - The article "the" is missing in several places.

---

> ### Author Response · Authors · 2022-08-02
> **Response**
>
> We thank the reviewer for the detailed comments, and will address them in order below.
>
> Questions:
> 1. Smoothing of the simplex may appear to make $K’$ nonconvex around the origin, but it should be noted that $K'$ is not actually the ball, but the intersection of the ball and {$\{u : u \ge \sqrt{\epsilon}\}$} for an $\epsilon$ smoothing of the simplex.  As an intersection of convex sets, this is convex. As such, quadratic reparameterization over the ball should still work. As noted by reviewer NwoT, there are other challenges involving the Lipschitz constants here, but sublinear regret is still attainable.  We leave improving rates to future work.
>
> 2. Theorem 7 is a preliminary attempt to characterize general conditions under which convergence of a non-convex OGD is possible. Unfortunately, we do not have any concrete applications of Theorem 7 beyond the settings of Theorem 3.
>
> Limitations: As noted above, we can still get sublinear regret for reparameterization of exponentiated gradient and similar smoothing and balancing of parameters should also suffice for a log-barrier.
>
> Typos: We have fixed the typos in the most recent revision. Thank you for pointing these out.

---

> > ### Comment · Reviewer_x2pL · 2022-08-06
> > **A quick question**
> >
> > Perhaps I misunderstand something: Isn't the set $\set{ u: u \geq \sqrt{\epsilon} }$ nonconvex?

---

> > > ### Author Response · Authors · 2022-08-07
> > > **Convexity of constraint set**
> > >
> > > Perhaps the notation is confusing.  More precisely, we mean $S =$ {$u | \forall i, u_i \geq \sqrt{\epsilon}$}.  This set is convex as for  $u, w \in S$, for all $\lambda \in [0,1]$, $\lambda u + (1-\lambda)w \in S$ because if $u_i \geq \sqrt{\epsilon}$ and $w_i \geq \sqrt{\epsilon}$ then $\lambda u_i + (1-\lambda) w_i \geq \sqrt{\epsilon}$. Hope this helps!

---

> > > > ### Comment · Reviewer_x2pL · 2022-08-07
> > > > **Thanks**
> > > >
> > > > Thanks. Sorry I had an illusion when imagining the shape of the set.

---

> > > > > ### Comment · Reviewer_x2pL · 2022-08-07
> > > > > **Appendix B & comment**
> > > > >
> > > > > I just saw your response to another reviewer that you worked out the EG example in Appendix B. The proof looks fine. There is a typo in Line 429: It is $\epsilon$ that is chosen as $1/23$.
> > > > >
> > > > > The idea in this paper is interesting and simple. Unfortunately, this paper does not provide any relevant application in which the regret of the online nonconvex optimization problem was unclear and can be analyzed by the framework of this paper. So I cannot give a high score.

---

> > > > > > ### Author Response · Authors · 2022-08-07
> > > > > > **Application vs. explaining the phenomenon**
> > > > > >
> > > > > > Dear reviewer,
> > > > > > We respectfully disagree about the non-convex regret being clear before us - getting a finite time regret bound was explicitly posed as an open question in the last NeurIPS by experts in the field, and as far as we know, we are the first to get finite time regret bounds!
> > > > > > (which as evident from other questions of the reviewers, require some calculations in various cases and not immediate).

---

> > > > > > > ### Comment · Reviewer_x2pL · 2022-08-07
> > > > > > > **Unclear relevance**
> > > > > > >
> > > > > > > This paper provides a candidate framework for online nonconvex optimization, but if this candidate cannot help us solve any specific problem of researchers' interest, then its relevance is in question. That's why I asked for at least one relevant application so the relevance can be justified.
> > > > > > >
> > > > > > > There should be already other candidate frameworks for online nonconvex optimization. For example, geodesic convexity is also one approach to handle nonconvex optimization. The paper "No-regret online learning over Riemannian manifolds" by Wang et al. provides regret guarantees and applications.

---

> > > > > > > > ### Author Response · Authors · 2022-08-07
> > > > > > > > **constrained vs. unconstrained  non convex optimization**
> > > > > > > >
> > > > > > > > Dear reviewer,
> > > > > > > >
> > > > > > > > We see your point, it would have been great to analyze a case in which GD converges to global optimality in nonconvex optimization that is completely new.
> > > > > > > >
> > > > > > > > 1. We do accomplish this partially, since we can handle constrained optimization with projections, which were not analyzed in the previous line of work, which analyzed unconstrained optimization.
> > > > > > > >
> > > > > > > > 2. We agree that coming up with a new case completely would be better, we still see value in precise analysis of the finite time regret bounds: this leads to better and more precise understanding of the limitations of the technique.

---

### Meta-Review · Area_Chair_KpiE · 2022-08-30

**Recommendation:** Accept
**Confidence:** Less certain

**Metareview:**

The main result of the paper is on establishing an approximate equivalence between online gradient descent (OGD) on non-convex losses with online mirror descent (OMD) on convex reparametrizations of the losses. In a previous result by Amid and Warmuth, which applies to the the continuous-time setting, we have exact conditions where the gradient flow with this reparameterization is exactly equivalent to continuous-time mirror descent. However, the current paper focuses on providing discrete time algorithms. The main theoretical contribution of the paper is to provide sufficient general conditions for the approximate equivalence to go through, leading to an O(T^{2/3}) regret bound for OGD on non-convex losses.

The paper was discussed to a good extent between the authors and the reviewers, and among the reviewers. The reviewers (and the AC) agreed that the paper provides a novel analysis which is based on a simple, elegant, and novel algorithmic equivalence method. Most of the concerns of the reviewers were addressed during the discussion phase. However, the reviewers still felt that the paper lacks relevant and important applications where the approximate equivalence (and reparametrization) would apply to and would lead to non-trivial results. All in all, the reviewers agreed on the decision that the paper lies on the acceptance border (with inclination towards acceptance).


**Award:**

No

---

### Decision · Program_Chairs · 2022-09-14

Accept